# A Systematic Evaluation of High-Throughput Sequencing Approaches to Identify Low-Frequency Single Nucleotide Variants in Viral Populations

**DOI:** 10.3390/v12101187

**Published:** 2020-10-20

**Authors:** David J. King, Graham Freimanis, Lidia Lasecka-Dykes, Amin Asfor, Paolo Ribeca, Ryan Waters, Donald P. King, Emma Laing

**Affiliations:** 1The Pirbright Institute, Woking, Surrey GU24 0NF, UK; dking1@dstl.gov.uk (D.J.K.); graham.freimanis@pirbright.ac.uk (G.F.); Lidia.Lasecka-Dykes@pirbright.ac.uk (L.L.-D.); amin.asfor@pirbright.ac.uk (A.A.); ryan.waters@pirbright.ac.uk (R.W.); donald.king@pirbright.ac.uk (D.P.K.); 2Department of Microbial and Cellular Sciences, Faculty of Health and Medical Sciences, School of Biosciences and Medicine, University of Surrey, Guildford GU2 7XH, UK; 3Department of Pathology and Infectious Diseases, Faculty of Health and Medical sciences, School of Veterinary Medicine, University of Surrey, Guilford GU2 7XH, UK; 4Biomathematics and Statistics Scotland, Edinburgh, Midlothian EH9 3FD, UK; pribeca@bioss.ac.uk

**Keywords:** high-throughput sequencing, viral populations, sub-consensus variants, sequencing error

## Abstract

High-throughput sequencing such as those provided by Illumina are an efficient way to understand sequence variation within viral populations. However, challenges exist in distinguishing process-introduced error from biological variance, which significantly impacts our ability to identify sub-consensus single-nucleotide variants (SNVs). Here we have taken a systematic approach to evaluate laboratory and bioinformatic pipelines to accurately identify low-frequency SNVs in viral populations. Artificial DNA and RNA “populations” were created by introducing known SNVs at predetermined frequencies into template nucleic acid before being sequenced on an Illumina MiSeq platform. These were used to assess the effects of abundance and starting input material type, technical replicates, read length and quality, short-read aligner, and percentage frequency thresholds on the ability to accurately call variants. Analyses revealed that the abundance and type of input nucleic acid had the greatest impact on the accuracy of SNV calling as measured by a micro-averaged Matthews correlation coefficient score, with DNA and high RNA inputs (10^7^ copies) allowing for variants to be called at a 0.2% frequency. Reduced input RNA (10^5^ copies) required more technical replicates to maintain accuracy, while low RNA inputs (10^3^ copies) suffered from consensus-level errors. Base errors identified at specific motifs identified in all technical replicates were also identified which can be excluded to further increase SNV calling accuracy. These findings indicate that samples with low RNA inputs should be excluded for SNV calling and reinforce the importance of optimising the technical and bioinformatics steps in pipelines that are used to accurately identify sequence variants.

## 1. Introduction

Advances in high-throughput sequencing (HTS) technologies such as those offered by Illumina allow for the rapid generation of large amounts of deep sequence data. These data can be used to identify sub-consensus single-nucleotide variants (SNVs) essential for understanding viral populations. Indeed, HTS technologies have been used to identify both common and rare SNVs [1] associated with: drug resistance [2,3], immune escape [4,5], and evolution and transmission pathways [6,7,8,9], with important implications for both human and animal health. Nevertheless, challenges still exist to distinguish real variation from process-introduced bias.

In order to obtain the requisite high genome coverage depth, polymerase chain reaction (PCR) amplification of the sequencing target is normally required [10], with reverse transcription (RT) being a prerequisite in the case of RNA sample to generate cDNA before amplification. While the RT step is a non-expansive process, PCR amplification is cumulative, and any technical errors produced at either step will be indistinguishable from true variation after subsequent PCR cycles [11]. The sequencing process also has to be considered, with a recent study declaring that the Illumina sequencer itself generates the most error [12]. In an attempt to circumvent these errors, different laboratory approaches and methodologies have been developed. For example, the use of high-fidelity enzymes [10] and tailored protocols including; CirSeq, a rolling-circle based RT of circularised RNA to generate repeated cDNA copies [13], and PrimerID, which involves the use of a degenerate block of nucleotides embedded into a RT primer allowing DNA to be tracked through the PCR and sequencing process [14]. However, both methods have limitations, with CirSeq requiring large amounts of starting nucleic acid template [13] and PrimerID suffering from reduced PCR amplification efficiency due to the formation of primer hairpins and/or dimers [15].

The bioinformatic pipelines to identify SNVs normally involves the alignment of filtered fastq files against a suitable reference sequence with an alignment software. The resulting Sequence Alignment Map (SAM) and Binary Alignment Map (BAM) files can be used to perform visual controls of the alignment, generate coverage plots, consensus sequences and to identify SNVs. This later point, along with viral haplotype reconstruction analysis can be limited from the accuracy of the dataset. These computational analysis steps employ specific parameters and algorithms that also influence SNV identification [16]. An increasing number of available programs, each claiming to offer the highest level of accuracy [17,18,19,20], can also make it difficult to select an algorithm that introduces the least amount of bias.

While previous studies have simultaneously and systematically investigated the impact of laboratory and bioinformatics protocols on SNV calling and error generation [21], to the best of our knowledge the focus of these previous studies has always been on intermediate coverage and high-frequency SNVs. Here, for the first time, we have systematically evaluated the impact of different laboratory and bioinformatic approaches in an HTS pipeline for identifying low-frequency SNVs in high-coverage viral datasets. Artificial RNA and DNA populations were used to differentiate the effects of distinct steps within the technical protocol. Following reverse transcription (for the RNA population) and PCR amplification, these samples were sequenced on an Illumina MiSeq platform providing data that were used to assess the accuracy of SNV detection through the generation of micro-averaged Matthews correlation coefficient (MuMCC) scores. Variant calling software such as LoFreq and VirVarSeq [22,23] apply their own criteria such as alignment quality, base quality and coverage to call SNVs [24]. Because of this, we have deliberately taken a simple, agnostic approach to assess the laboratory and bioinformatic pipelines on calling SNVs without dependency on a variant callers assumptions of data distribution. Assessing multiple possible combinations and parameters, we were able to determine the key factors that influence the overall performance of an HTS pipeline, considering the type and amount of input material and to identify systematic patterns of error.

## 2. Materials and Methods

An overview of the laboratory and computational pipelines is shown in Figure 1.

### 2.1. Preparation of DNA and RNA Populations

The starting template for DNA and RNA populations were derived from five pT7S3 plasmids (named wild type, A, B, C and D), each 11,278 bp in length and containing a full-length genome insert of O_1_Kaufbeuren strain of foot-and-mouth disease virus (FMDV) (8218 bp) (GenBank: EU448369) [25]. Site-directed mutagenesis was performed on four of these plasmids (pT7S3 A–D) to introduce substitutions at known positions in the capsid-encoding region of the virus (Table 1). Each plasmid was then transformed into separate JM109 *E. coli* competent cells (Promega, Southampton, UK) by a standard heat-shock protocol, with cells being incubated separately overnight at 37 °C on agar plates (imMedia™ Growth Medium (Themo Fisher Scientific, Massachusetts, USA)). Single colonies were selected and used to inoculate individual 20 mL Luria-Bertani (LB) broths (The Pirbright Institute, Surrey, UK). After incubation for 16 h at 37 °C at 225 rpm, each of the five plasmids were purified using the QIAprep mini kit (Qiagen, Crawley, UK), as per manufacturer’s instructions, with sanger sequencing used to confirm the presence of the nucleotide substitutions and the Nanodrop used to estimate the DNA yield [26].

Plasmids were linearised using *HpAI* (New England Biolabs, Hertfordshire, UK) as per manufacturer’s instructions. Using the MEGAScript T7 kit (ThemoFisher Scientific), RNA was transcribed from 1 µg of each of the linearised plasmids, with TURBO DNase and the MEGAclear transcription clean-up kit (Thermal Fisher Scientific) being used to remove input plasmid DNA (as per manufacturer’s instructions). A qRT-PCR assay, with and without the RT enzyme, demonstrated that the synthesised product was >99.99% RNA [26] Artificial DNA and RNA populations were created by mixing the individual plasmids or in vitro transcripts at defined ratios (wild type: 0.01%, A: 1%, B: 10%, C: 88.89% and D: 0.1%) to achieve different nucleotide frequencies (Table 1): 11% frequency (*n* = 2), 1.11% frequency (*n* = 2), 0.11% frequency (*n* = 8) and 0.01% frequency (*n* = 8).

In order to mimic real samples, bovine genomic material was used to dilute the DNA and RNA populations. For this, 25 mg of bovine tongue epithelium tissue (collected from Newman’s Abattoir, Farnborough, UK), was added to 500 µL of TRIzol reagent (Thermo Fisher Scientific, Massachusetts, USA) in a 2 mL microcentrifuge tube containing a single 5 mm steel bead (Qiagen, Crawley, UK). Tubes were placed in the TissueLyser LT instrument (Qiagen, Crawley, UK) before the epithelium tissue was homogenised at 20 oscillations per second for two minutes. This was followed by a 5 min incubation at 4 °C and centrifugation for 10 min at 8000× *g*. Following this, supernatant was then transferred into a Phasemaker tube (Thermo Fisher Scientific, Massachusetts, USA) with an additional 1.25 mL of TRIzol reagent and 300 µL of chloroform being added where after nucleic acid was extracted from the upper aqueous phase, as per the manufacturer’s instructions. Extracted nucleic acid template was confirmed FMDV negative via qRT-PCR [26].

Both DNA and RNA populations were diluted in the bovine genomic material to create three different starting inputs for RNA (*High* (10^6^ RNA copies/µL), *Medium* (10^4^ RNA copies/µL) and *Low* (10^2^ RNA copies/µL)) and two starting inputs for the DNA (*High* (10^6^ DNA copies/µL) and *Low* (10^2^ DNA copies/µL)). A total of 10 µL of each of the RNA populations (total copies: 10^7^ for RNA *High*, 10^5^ for RNA *Medium* and 10^3^ for RNA *Low*) were used in the Transcriptor High Fidelity cDNA Synthesis Kit (Roche, Welwyn Garden City, UK) in order to convert the RNA into cDNA. For this reaction, the manufacturer’s instructions were used with the addition of 2 µM of an oligo dT primer (REV6 [27], GGC GGC CGC TTT TTT TTT TTT TTT).

### 2.2. Polymerase Chain Reaction (PCR) Optimisation

PCR cycling conditions were optimised in order to both limit amplification bias and to produce the required input amount for the Nextera XT DNA library preparation kit (Illumina, San Diego, CA, USA) (0.2 ng/µL). Firstly, a 3022 bp product consisting of the leader and capsid encoding regions of FMDV was amplified from each population by using Platinum SuperFi DNA Polymerase (Thermo Fisher Scientific, Massachusetts, USA), as per the manufacturer’s instructions, with the addition of 3 µL of DNA of each population type and 10 µM of serotype universal FMDV capsid primers [28] (Forward: TGG TGA CAG GCT AAG GAT G (Genbank: EU448369 base position: 914) and Reverse: GCC CAG GGT TGG ACT C (pT7S3 base position: 3936)). Cycling conditions were as follows: 98 °C for 30 s, followed by 39 cycles of 98 °C for 15 s, 66 °C for 15 s and 72 °C for 2 min with a final cycling step of 72 °C for 5 min.

Aliquots were removed every 2 cycles (from 0 to 40), purified using the Illustra GFX PCR DNA and Gel Band Purification Kit (GE Healthcare, Little Chalfont, UK) as per the manufacturer’s instructions and eluted in 50 µL of nuclease-free water, prior to quantification using the Qubit^®^ dsDNA High Sensitivity Assay Kit (Thermo Fisher Scientific, Massachusetts, USA). Results were used to determine a PCR cycle number for each DNA and RNA population input.

### 2.3. Illumina Sequencing

Following PCR optimisation, all RNA and DNA inputs sets were processed in quadruplicates (with the RT and PCR steps being performed independently). Samples were diluted to 0.2 ng/µL in nuclease-free water prior to library preparation using the Nextera XT DNA sample preparation kit. Final libraries were multiplexed and diluted to 12.5 pM prior to sequencing on an Illumina MiSeq platform using a single 2 × 150 cycle, paired-end sequencing run using a version 2 chemistry MiSeq reagent cartridge. Raw reads were deposited into GenBank under the BioProject accession number PRJNA669475.

### 2.4. Bioinformatic Analysis

After sequencing, quality control checks were performed on the raw fastq data using FastQC [29] (version 0.11.5). The first 15 and final 5 bases of each read were of lower average qScore quality compared to the rest of the read [26] and consequently removed from each read using Prinseq-lite (version 0.20.4) [30]. This produced an average read length 131 bp in length.

#### 2.4.1 Sequence Alignment and Assessment of High-Throughput Sequencing (HTS) Pipeline Performance

Trimmed reads were passed through an in-house bash (bash version 4.1.2(1)) script that iteratively (for each parameter assessed) processed reads through Sickle (version 1.33) [31], which filtered the reads ends according to quality (q0, q10, q20, q30, q35 and q38) and length (70, 80, 90, 100, 110, 120 and 130). Following this, reads were independently aligned to a reference genome (pT7S3, GenBank: EU448369) using one of five different commonly used short read aligners; BWA-MEM (version 0.7.12-r1039) [19], GEM3 (version 3.6-2-g77d1) [20], Bowtie2 (version 4.1.2) [18] (both local and global alignment), HiSAT2 (version 4.8.2) [17] and SMALT (version 0.7.6) [32]. For each set of quality and length filtered reads, different sets of parameters of each aligner were used. A full list of parameters can be found in Appendix A. Following read mapping of each tested aligner and parameter, BEDTools [33] was used to calculate the mean coverage across the amplicon.

Using an in-house R-script (RStudio, R version 3.3.1) each alignment file was processed to allocate, for each sequenced position, and all tested frequency thresholds (Appendix A), the base call into 1 of 4 classes: 1. True Positive (TP), a known SNV called as a SNV, 2. True Negative (TN), a non-SNV site with no SNV called, 3. False Positive (FP), a SNV called at a non-SNV site, i.e., an error, and 4. False Negative (FN), no SNV called at positions where a true SNV was present. 

The calling of a SNV was dependent on the frequency threshold being applied, with SNVs identified below a tested threshold being considered as a TN, for example, at a frequency threshold of 0.5%, known-SNVs at 0.01% were considered as a TN. This approach was subsequently repeated for all possible duplicate (*n* = 6), triplicate (*n* = 4) and quadruplicate (*n* = 1) combinations. For each replicate combination, the classification of each position was based on 100% agreement, i.e., a TP was assigned if all samples in a combination agree. Finally, the TP, TN, FP and FN values across all combinations of replicates over all bases positions were used to calculate a micro-averaged MuMCC, a measure of classification performance, which unlike sensitivity and specificity tests, considers the entire confusion matrix and is able to handle an imbalance of classes (i.e., few true SNVs), for each base position:(1)$$MuMCC=\frac{\sum_{c}TP_{c}\sum_{c}TN_{c}-\sum_{c}FP_{c}\sum_{c}FN_{c}}{\sqrt{\left(\sum_{c}TP_{c}+\sum_{c}FP_{c}\right)\left(\sum_{c}TP_{c}+\sum_{c}FN_{c}\right)\left(\sum_{c}TN_{c}+\sum_{c}FP_{c}\right)\left(\sum_{c}TN+\sum_{c}FN_{c}\right)}}$$
(i.e., micro-average all singlets, micro-average of all duplicate combinations, micro-average of all triplicate combinations, micro-average of quadruplicate combination). A perfect correlation between the predicted and observed, equivalent to 100% accuracy in SNV calling, was indicated by an MuMCC score of 1, randomness of the data by an MuMCC score of 0 and a total discordance by an MuMCC score of −1.

The MuMCC across all sequence positions was used to represent the average performance of a candidate HTS pipeline comprising a combination of experimental and bioinformatic factors (aligners, replicate number and frequency threshold) from which the performance assessment was derived. The effect of each laboratory and bioinformatic parameter on MuMCC performance was assessed using ANOVA (R (version: 3.3.1), aov). Tukey’s honest significant different (HSD) test (R (version: 3.3.1), Tukey HSD) was performed post-hoc for parameters identified as having a significant effect (*p* value ≤ 0.01). Following these tests, the aligner, qScore, read length and percentage frequency threshold parameters that produced the highest MuMCC scores were selected in turn for each population type. Areas with higher percentages of error were identified and were used to investigate sequence patterns along the template which could contribute to systematic error.

## 3. Results

### 3.1. PCR Cycle Optimisation and Illumina Sequencing

A minimum number of PCR cycles required to produce enough material for the Nexteria XT protocol (1ng) for each population type was established. A total of 18 and 34 cycles was found to be the optimal number for DNA *High* and *Low* inputs respectively. While for the RNA populations, the total number of PCR cycles required was 26, 34 and 40 for *High*, *Medium* and *Low* inputs, respectively.

Following sequencing, a total of 2.52 × 10^7^ reads were produced, with a mean of 1.26 × 10^6^ reads generated for each sample. Further information regarding the statistics of the MiSeq sequencing run can be found in Appendix A.

#### 3.1.1 Coverage

Following sample alignment, BEDTools was used to calculate the average coverage across the amplicon for all parameters tested (aligners, aligner parameters, read parameters and cut-off thresholds). For the DNA *High* sample set, mean coverage ranged from 2.33 × 10^3^ (Bowtie2-Global alignment on replicate 2) to 7.05 × 10^4^ (BWA-MEM, Bowtie2-Local and SMALT alignment on replicate 4), while for the DNA *Low* samples, mean coverage ranged from 3.51 × 10^3^ (HiSAT2 alignment on replicate 4) to 6.63 × 10^4^ (Bowtie2-Local alignment on replicate 4).

For the RNA *High* samples, mean coverage ranged from 1.60 × 10^3^ (Bowtie2-Global alignment on replicate 4) to 6.52 × 10^4^ (BWA-MEN and Bowtie2-Local alignment on replicate 2), while for RNA *Medium* mean coverage ranged from 5.30 × 10^2^ (Bowtie2-Global alignment on replicate 4) to 6.14 × 10^4^ (BWA-MEM alignment on replicate 2). Mean coverage across the sequenced amplicon for the RNA *Low* samples ranged from 3.19 × 10^3^ (Bowtie2-Global alignment on replicate 4) to 6.63 × 104 (SMALT alignment on replicate 2).

Further details on mean sample coverage can be found in Appendix A.

### 3.2. The Effect of Input Nucleic Acid and Aligner Choice on the Accuracy of Single-Nucleotide Variant (SNV) Calling

To assess which of the tested aligners had the greatest influence on the accuracy of SNV calling, the range of MuMCC scores across all parameters tested (aligners, aligner parameters, read parameters and cut-off thresholds) was compared. Figure 2 shows that for all aligners tested, the DNA population had a greater maximum MuMCC, compared to their equivalent RNA population (Appendix A).

There was a positive relationship between the amount of starting input material for both DNA and RNA populations and the range of MuMCC scores, with a greater effect on performance observed for RNA populations. Indeed, a Tukey HSD test comparing the distributions of MuMCC between input material types showed a small (~0.1 change in correlation coefficient), but significant differences between DNA and RNA populations (RNA-DNA difference of MuMCC: −0.088, Tukey HSD *p* value ≤ 0.01). Performing the equivalent comparison for the abundance of material (i.e., *high*, *medium* and *low*) highlighted significant differences (Tukey HSD *p* value ≤ 0.01 between the levels of input material. With reduced genomics inputs, the overall difference in MuMCC scores dropped 0.083 between DNA *High* and DNA *Low*, 0.050 between RNA *High* and RNA *Medium* and 0.165 between RNA *Medium* and RNA *Low*.

Comparing across aligners, GEM3 produced the highest maximum MuMCC for RNA *High* and DNA *Low* (0.855 and 0.729 MuMCCs, respectively). An equal MuMCC was observed for all aligners (apart from Bowtie2-Global), for the RNA *Medium*, RNA *Low* and DNA *High* starting inputs (0.707, 0.250 and 1.00 MuMCCs respectively). Overall, Bowtie2 Global yielded the poorest MuMCC across all population sets tested (Appendix A). Post-hoc tests showed that the use of GEM3 as an aligner for all population types (except RNA *Low*) produced a significant increase (Tukey HSD *p* value ≤ 0.01) in accuracy against all aligners with the exception of BWA-MEM.

Although no overall significant differences were found between GEM3 and BWA-MEM (with the exception of RNA *Low*), GEM3 was selected on which all downstream analyses were to be based as it produced the highest MuMCC score for the RNA *High* input and overall mean MuMCC scores across all other inputs (Figure 2 and Appendix A).

### 3.3. The Effect of Replicate Combinations and qScore Choice on the Accuracy of SNV Calling

Having set the aligner to GEM3, the impact of using Sickle to trim the read ends based on a qScore threshold (q0, q10, q20, q30, q35 and q38) and the number of technical replicates on the overall accuracy of SNV calling was investigated. Figure 3 shows that using more than one technical replicate, had a positive effect on the overall MuMCC score, with the exception of the RNA *Low* input. Further information about the range of MuMCC scores for all population inputs can be found in Appendix A, while a full list of qScore parameters producing the highest accuracy for SNV detection can be found on Table 2.

#### 3.3.1. RNA Input

Comparing the distributions for the RNA *High* input found that a qScore of q38 for all technical replicates combinations produced the maximum MuMCC scores (0.855, 0.816, 0.894 and 0.894 for singlets, duplicate, triplicate and quadruplicates respectively). While for the RNA *Medium* input, a maximum MuMCC score of 0.707 was identified when using q35 for singlet and q38 for all other technical replicates (MuMCC difference between q35 and q38 for singlets was 0.015). 

Post-hoc analysis revealed that the MuMCC scores produced using a qScore of 38 across all technical replicate combinations in the RNA *High* input were significantly (Tukey HSD *p* value ≤ 0.01) higher than the mean MuMCC values of other qScore thresholds tested. This was also observed for the RNA *Medium* input. For the RNA *Low* input, the highest MuMCC score was obtained using a qScore of 35 (MuMCC: 0.279), while all other replicate combinations produced MuMCC scores between 0.001 and 0.005.

Based on the above, a qScore of q38 was selected for all replicate combinations of RNA *High* and *medium*. A qScore of q35 could be selected RNA *Low* when using singlet replicates only.

#### 3.3.2. DNA Input

For the DNA *High*, the maximum MuMCC score of 1 was identified when a qScore either q30, q35 or q38 was used for singlet and a qScore of q30 was used for duplicate, triplicate or quadruplicate technical replicates. For the DNA *Low* input, the highest MuMCC score of 0.707 was achieved at q35 for singlet and duplicate and q38 for triplicate and quadruplicate technical replicates. Tukey HSD post-hoc analysis indicated that the use of q38 for singlet and duplicate and q35 for triplicate and quadruplicate technical replicates produce significantly higher (Tukey HSD *p* value ≤ 0.01) results for the DNA *High* input compared to other qScore thresholds. For the DNA *Low* input, a qScore of 35 for all replicate combinations was the most significant (Tukey HSD *p* value ≤ 0.01). 

Based on the above results, a qScore of q35 was selected for all DNA populations and replicate combinations.

### 3.4. The Effect of Replicate Combinations and Read Length Choice on the Accuracy of SNV Calling

Following the choice of aligner and qScore threshold (from Table 2) the influence of read length and the number of technical replicates on SNV calling accuracy was tested. Although 151 sequencing cycles were used for each paired end, the read length profile of each sample set varied as a result of the adaptor and quality trimming procedures. The read length parameter, therefore, sets the minimum length of the trimmed reads. The distribution of the MuMCC scores of all tested read lengths for all population inputs can be found on Appendix A.

Analysis found that either multiple or no significant read length parameters were identified for all population inputs and replicate numbers. Therefore the choice of read length was based on the shortest length which gave the highest MuMCC scores (Table 2).

#### 3.4.1. RNA Input 

Using the RNA *High* input, a maximum MuMCC score of 0.816 was achieved for singlet (read length of 70–90 bp and 130 bp) and duplicate (all read lengths tested) replicate combinations. The maximum MuMCC score was found to increase to 0.894 for triplicate and quadruplicate technical replicates when a read length of 130 bp was used. However, minor differences of MuMCC values between the use of a 70 bp and 130 bp read length were identified for triplicates and quadruplicate technical replicates (0.011 and 0.009 MuMCC respectively).

In contrast, for the RNA *Medium* input, a maximum MuMCC score of 0.707 was identified regardless of read length and technical replicate input. While for the RNA *Low* input, only singlet replicates produced a mean MuMCC score above 0.01, with the highest being 0.149 when a read length of 100 and above was used. However, the difference in MuMCC score for RNA *Low* single replicates was 0.001 between a read length of 70 bp and 100 bp.

Due to the very small MuMCC differences (<0.011) between tested parameters, a read length of 70 bp was selected for all RNA populations and replicate combinations. 

#### 3.4.2. DNA Input 

In contrast, for DNA *High* input, a maximum MuMCC score of 1 was identified for read lengths of 70 bp, 100 bp, 110 bp and 130 bps. MuMCC scores were found to decrease with the addition of technical replicates, with a maximum score of 0.933 found with duplicate (70 bp, 110 bp, 130 bps) and 0.913 for triplicate (100 bp) and quadruplicate (70 bp, 80 bp, 90 bp, 100 bp) replicate combinations respectively. The MuMCC difference between all read lengths tested was found to be <0.01. 

For the DNA *Low* a maximum MuMCC score of 0.707 was identified for all technical replicate combinations regardless of the read length used.

Based on the above results, a read length of 70 bp was selected for all DNA replicate combinations

### 3.5. How Does Frequency Cut-Off Impact the Accuracy of Variant Calling?

Using the GEM3 aligner and fixing the qScore and read length parameters (to the values indicated in Table 2), the influence of each percentage frequency threshold between 0.01% and 1% (for SNV detection) and the number of technical replicates on the HTS pipeline performance (as measured by the highest MuMCC score) was assessed. For each differing GEM3 alignment parameter tested (mapping mode (fast and sensitive) and maximum alignment error (0.05, 0.10, 0.12, 0.15)) for each tested frequency threshold (Appendix A), a single MuMCC value was produced. These results indicate that the GEM3 alignment parameters have no effect on SNV detection performance. Following this, the MuMCC value from each replicate number was subsequently visualised to identify an optimal frequency threshold (Figure 4).

Overall, MuMCC score analysis showed that a greater abundance of starting material and a higher number of technical replicates (except DNA *Low* and RNA *Low*) allowed for a reduced percentage frequency cut-off to be applied for SNV calling. Further information regarding each frequency threshold can be found within Appendix A and a list of frequency thresholds which produced the most accurate MuMCC scores can be found in Table 2.

#### 3.5.1. RNA Input

Within the RNA *High* input, the maximum MuMCC score obtained was at a frequency of 0.2% for all replicate combinations, with MuMCC scores increasing from 0.756 for singlets to 0.816 for duplicate replicate combinations and above. For percentage thresholds above 0.2%, the MuMCC scores decreased due to the reduced number of TPs after 0.l% threshold. Below 0.2%, other high MuMCC scores were observed within the duplicate, triplicate and quadruplicate data at 0.06% (MuMCC: 0.535, 0.632 and 0.680 respectively), suggesting this frequency could be applied to characterise SNVs below 0.2% at the cost of allowing more errors through the analysis process. 

For the RNA *Medium* input, the advantage of increased technical replicates was clear, with a percentage frequency cut-off of 0.8% identified as the most accurate frequency threshold for singlets, while 0.5% was identified for duplicates, 0.3% for triplicates and 0.2% for quadruplicates (MuMCC: 0.707 for all replicate numbers).

For the RNA *Low* input, only the singlet dataset produced MuMCC scores above 0.01, with the highest MuMCC score obtained from 0.5% frequency cut-off onwards (MuMCC: 0.250). The MuMCC scores for duplicate, triplicate and quadruplicate combinations ranged from −0.003 to 0.007, implying that these results were no better than random. This and the low MuMCC score for singlets demonstrates that the RNA *Low* input material cannot be used for accurate SNV analysis. 

#### 3.5.2. DNA Input

With the DNA *High* input, the maximum MuMCC score obtained for singlet and duplicate replicate combinations was achieved using a frequency cut-off threshold of 0.2% (MuMCC: 1.000 and 0.933 respectively). As with the RNA input, another high MuMCC score was observed at 0.05% for singlet and 0.04% for duplicate combinations (MuMCC: 0.764 and 0.840 respectively), suggesting that this frequency could be applied to characterise SNVs at the cost of allowing more errors through the analysis process. For triplicate and quadruplicate technical replicates, a frequency threshold of 0.04% produced the highest MuMCC scores (MuMCC: 0.891 and 0.913 respectively).

For the DNA *Low* input, a maximum MuMCC score of 0.707 was achieved at a 0.2% frequency regardless of replicate combinations. 

### 3.6. False Positive Patterns and Distrubutions 

Once the optimised conditions for the HTS SNV calling pipeline had been established (to the values indicated in Table 2), the patterns and distributions of FPs were assessed to identify systematic patterns of error. Firstly, the type of FP (regardless of frequency cut-off thresholds) for each replicate sequenced from each population type was investigated. All population types showed the same pattern of transition FPs; T to C base change, A to G base change, C to T base change and G to A base change. Each of these four base changes represented >10% of all FP types (Figure 5). After identifying the error type, FPs found above the recommended frequency threshold for each replicate sequenced for each population type were investigated. 

Within the RNA *High* input, a total of 4 FPs were identified above the 0.2% frequency cut-off, all of which were T to C base changes (Table 3). FPs at positions 1135 and 3056, were found to occur in all replicates and corresponded to the start of a 5-mer homopolymeric T region, while the other two FPs were present at the start of a 3-mer homopolymeric T region. While for the RNA *Medium* input dataset, only one FP above the 0.8% frequency cut-off was identified in one technical replicate, which was a T to C base change (Table 3) and corresponded to the start of a 5-mer homopolymeric T region. While no recommended frequency cut-off parameters below 1% were identified for the RNA *Low* input, five high frequency FPs, including two consensus level changes, unique to a single replicate were identified. Three were T to C base changes, a fourth a G to T base change and a fifth a G to C base change (Table 3). Overall, he highest frequency FPs occurring above the recommended cut-off threshold for the RNA population were T to C transitions occurring at the start of a homopolymeric region of T bases, with a higher number of T bases resulting in a higher frequency error.

For the DNA *High* input, no errors were found in any replicate above the recommended frequency cut-off of 0.2%. Whilst for the DNA *Low* input dataset, a single error G to T base change was identified above the 0.2% frequency threshold in replicate 4 (Table 3).

## 4. Discussion

The ability of HTS technologies to identify low-frequency SNVs is still limited by the presence of process-introduced errors that can mask true biological variation. Here, a systematic approach was taken to evaluate both laboratory and bioinformatics protocols to define an HTS pipeline(s) able to most accurately identify true biological variation. 

By investigating different DNA and RNA starting inputs, it was evident that the type and abundance of nucleic acid template impacted SNV call accuracy. As expected, fewer errors were present in the DNA populations, with error decreasing as more starting template was included (Figure 2). As only 0.2 ng/µL of DNA was required by the Nextera XT kit for sequencing on the Illumina MiSeq, the number of PCR cycles of each population type and abundance was optimised to produce enough material, whilst limiting the number of cycles. The increased number of PCR cycles required to accommodate lower amounts of starting template is thought to lead to the of presence of skewed allelic frequencies through preferential amplification of a small number of SNVs [34]. This was seen most clearly in the RNA *Low* input, with two out of the four technical replicates having consensus-level error. The increased frequency of errors in the RNA dataset can also be explained by those introduced by the RT step [11] and the conversion efficiency of the RT enzyme used, with one study finding the conversion efficiency of SuperScript III (Invitrogen) on bacteriophage MS2 RNA being between 35% and 69% [35]. Inefficient conversion of RNA to cDNA, or low amounts of starting RNA template may increase the likelihood of preferential amplification of SNVs and/or errors during PCR, as reduced cDNA will increase the likelihood of the same genome coming in contact with the polymerase more than once. As a result of these factors, the highest amount of available starting template for both DNA and RNA is required to maximise SNV detection accuracy.

We also show that specific algorithms behind short-read alignment programs can influence SNV calling. Five different aligner that each employ different algorithms were tested in order to maximise the accuracy of candidate HTS pipelines. This study found that both GEM3 and BWA-MEM for all samples types produced significantly accurate results (Tukey HSD *p* value ≤ 0.01) compared to the other tested aligners, with GEM3 producing the highest mean MuMCC score. Because of this, GEM3 was selected as the aligner on which to base all subsequent investigations (although BWA-MEM with specific parameter combinations could produce equivalent or better results). The next step was to select the number of replicates required, the read qScore and length that would be passed to GEM3, as well as the frequency cut-off required for SNV calling. A previous study investigating the role of technical and biological replicates for HTS for negating errors caused by the sample preparation and sequencing found that replicates can be used to filter error, increasing the confidence in low-frequency SNVs being called [36]. However, in this study the opposite was true for RNA *Low* input, with more than one technical replicate resulting in a MuMCC score of almost 0, suggesting that these results were no better than random. A possible explanation for this observation could be due to the low number of starting genomes, with only a small subsection of the RNA population being sequenced in each replicate, reducing the overlap of SNV calls between replicates. Due to the low MuMCC scores, high frequency and consensus level errors present and the fact that the use of technical replicates decrease confidence in the data, it is suggested that sequencing RNA at the same copy number as RNA *Low* (10^2^ RNA copies/µL) on an Illumina MiSeq platform set using the HTS pipelines in this study be avoided for analysing low-frequency SNVs.

Whilst single replicates could be applied for all sample types to predict low-frequency SNVs, this study shows that the use of duplicate technical replicates did increase the accuracy of SNV calling (as measured by MuMCC scores) for the RNA *High* input. The use of more than one technical replicate did not increase the MuMCC score for the RNA *Medium* and DNA *Low* inputs (maximum MuMCC: 0.707 for both), while the MuMCC score was found to decrease with additional DNA *High* (from 1 to 0.933) and RNA *Low* replicates (from 0.250 to ~0). However, this study showed that for RNA *Medium* and DNA *High* inputs, additional technical replicates allowed for SNVs to be called at lower percentage frequencies (Table 2) whist maintaining maximum accuracy. In previous studies, where a percentage frequency cut-off has been used to identify SNVs, no consideration for input material or PCR amplification was applied to reduce processed-introduced errors being called as real SNVs [8,37,38]. This study highlights the need for a tailored approach to frequency thresholds depending on template input concentration. A frequency cut-off of 0.2% was found to be the lowest point at which variations could be called with the highest accuracy (with the RNA *Medium* data requiring 4 replicates for this frequency cut-off), with the exception of the DNA *High* input, where the use of triplicate and quadruplicate combinations allowed for a frequency cut-off as low as 0.04%, which is close to the intrinsic sequencing limit of Illumina chemistry [12]. The use of more than one technical replicate (Figure 4) for the RNA *High* and DNA *High* inputs saw a peak in the MuMCC data at frequency thresholds of 0.06% and below (Table 2 and Appendix A). This suggests that these percentage frequency thresholds could be applied at the risk of allowing more errors through the bioinformatics pipeline. Future studies, however, could improve on this frequency cut-off threshold as the limited number of TP within the artificial populations may have impacted the accuracy of the results.

A study investigating error rates in just Illumina sequencing found that errors were more likely to occur in repetitive regions; however, at the end of the respective region and T > C base change errors were least likely to occur [39], which suggest, that the RT and/or PCR steps have a larger influence on error. Another sequencing study that utilised the same pT7S3 FMDV-O plasmid used here, found that when just PCR was used, that T to C transitions were one of the most abundant errors (along with A to G). This effect was also observed when the plasmid RNA was reverse transcribed and amplified [11]. Although the same error patterns were observed within the DNA and RNA datasets, greater frequencies of error occurred within the RNA populations, presumably due to the RT step. The use of RT within the RNA *High* input dataset resulted in systematic error from a T to a C base at the start of repetitive T base regions, above the chosen 0.2% frequency cut-off for RNA *High*. As a result, it is recommended that low-frequency T > C based SNVs are treated with caution or removed, The frequencies of predictable error maybe influenced by certain factors, for example: length of homopolymeric region, reagents, sequencing technologies used and individual users of the RT, PCR and sequencing steps. Interestingly, this reproducibility of error within technical replicates is only seen within the RNA *High* input and, as the RNA input template is reduced, the number of errors identified at similar percentage frequencies in more than one replicate decreases; however, further work is required to investigate this.

## 5. Conclusions

In this study, we demonstrated that different nucleic acid types and starting inputs have the greatest impact on the accuracy of calling sub-consensus SNVs. The use of the highest amount of starting template (≥1 × 10^6^ copies/µL), coupled with the use of more than one technical replicate can lead to the detection of SNVs ≥0.2% in frequency (and as low as 0.04% dependent upon the sample input type and number of technical replicates used).

Furthermore, the use of an RT step on a sample with a high RNA copy number (10^6^ RNA copies/µL) can lead to higher frequencies of predictable error (T to C base error patterns at the start of homopolymeric T base region), which can be used to further exclude error from real variation. We also identified that the use of low RNA inputs (10^2^ RNA copies/µL) led to the presences of high-frequency and consensus-level errors and the use of more than one technical replicate decreased MuMCC scores to ~0, indicating that SNVs called here were no better than random. Our recommendation is that that low RNA inputs should not be used for SNV calling.

By systematically characterising the laboratory and computational input factors, we have established a HTS framework which can be applied to both DNA and RNA viral populations at different inputs to accurately characterise low-frequency SNVs in high coverage Illumina datasets. As our results do not depend on sample origin, this framework can be generally applied to further understand viral population dynamics.

## Figures and Tables

**Figure 1 viruses-12-01187-f001:**
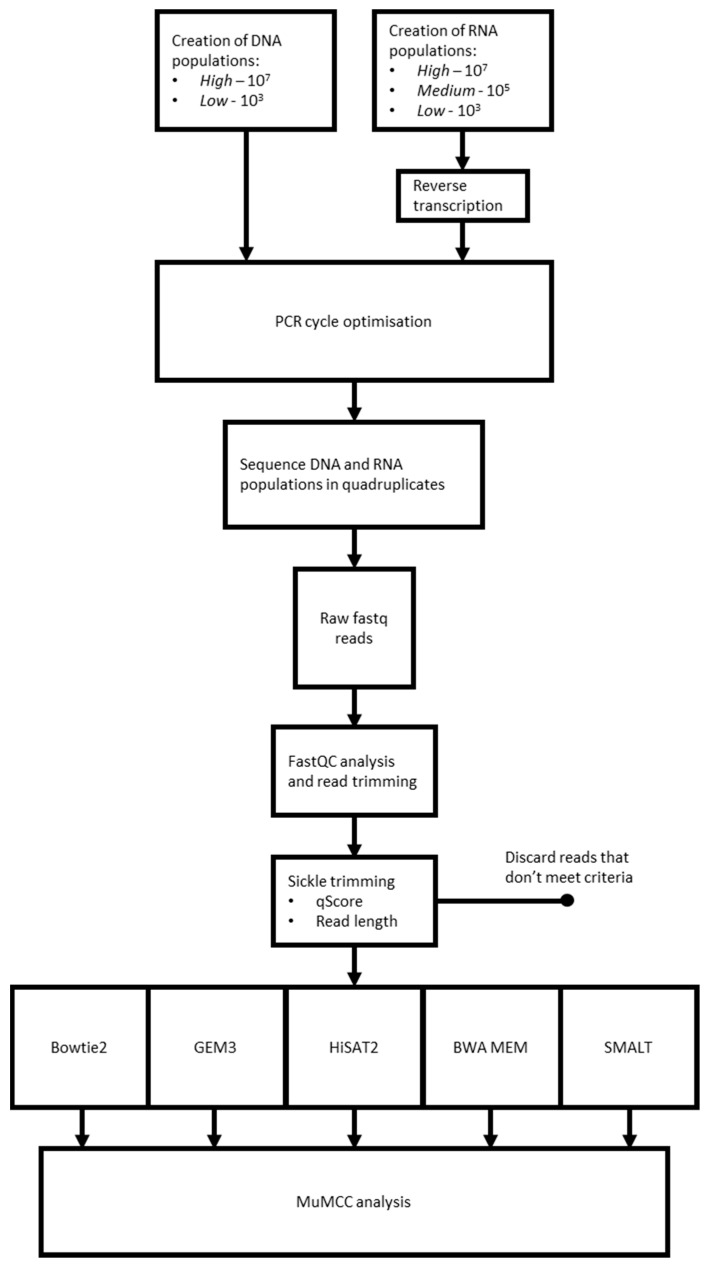
An overview of the high-throughput sequencing (HTS) pipelines evaluated in this study. Each pipeline comprised different combinations of laboratory and computational approaches.

**Figure 2 viruses-12-01187-f002:**
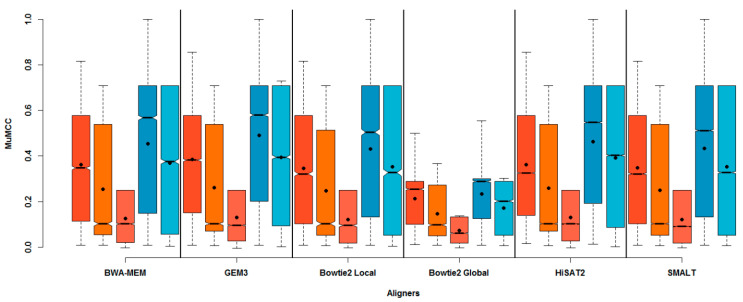
The effect of short read aligner choice on SNV calling accuracy. The range of micro-averaged Matthews correlation coefficient (MuMCC) scores of each of the tested aligners (including all parameters) used for all singlet population types. The mean of the MuMCC range is indicated by the solid black dot within each boxplot. Minimum and maximum whiskers on each bar plot indicate the highest and lowest MuMCC scores. RNA *High* = Red, RNA *Medium* = Orange, RNA *Low* = Light orange, DNA *High* = Blue, DNA *Low* = Light blue.

**Figure 3 viruses-12-01187-f003:**
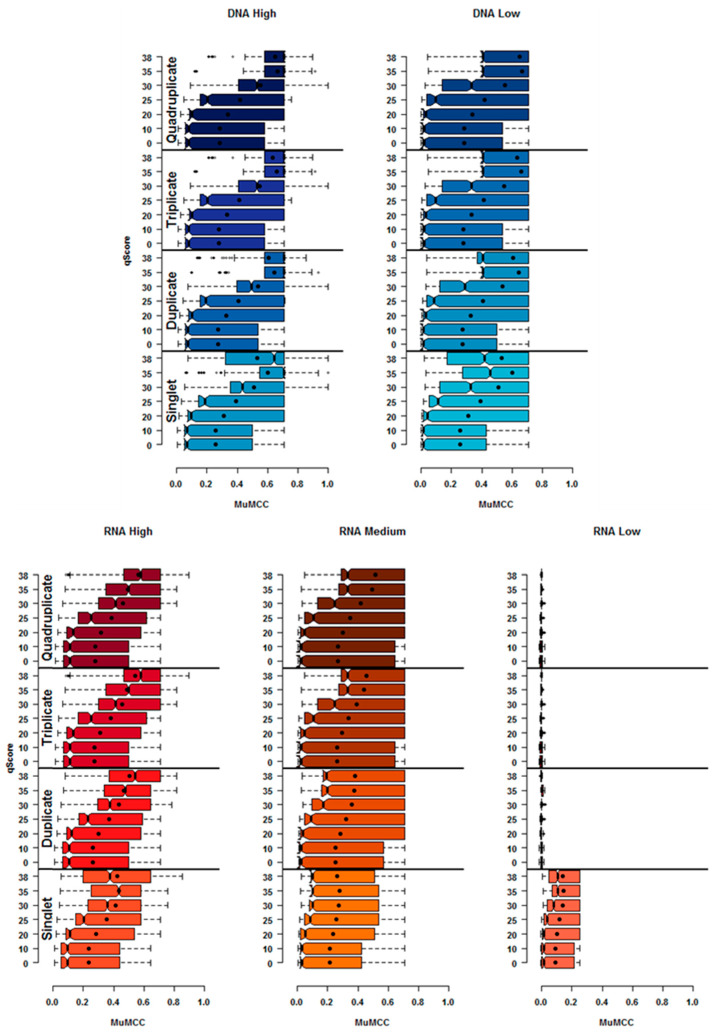
The effect of qScore and number of replicates on SNV calling accuracy. The range of MuMCC scores of each population input for each qScore parameter and technical replicate combinations tested. Singlet, duplicate, triplicate and quadruplicate technical replicates are represented by a different shade of colour. The solid black dot within each boxplot indicates the mean of the MuMCC distribution. Minimum and maximum whiskers on each bar plot indicate the highest and lowest MuMCC scores.

**Figure 4 viruses-12-01187-f004:**
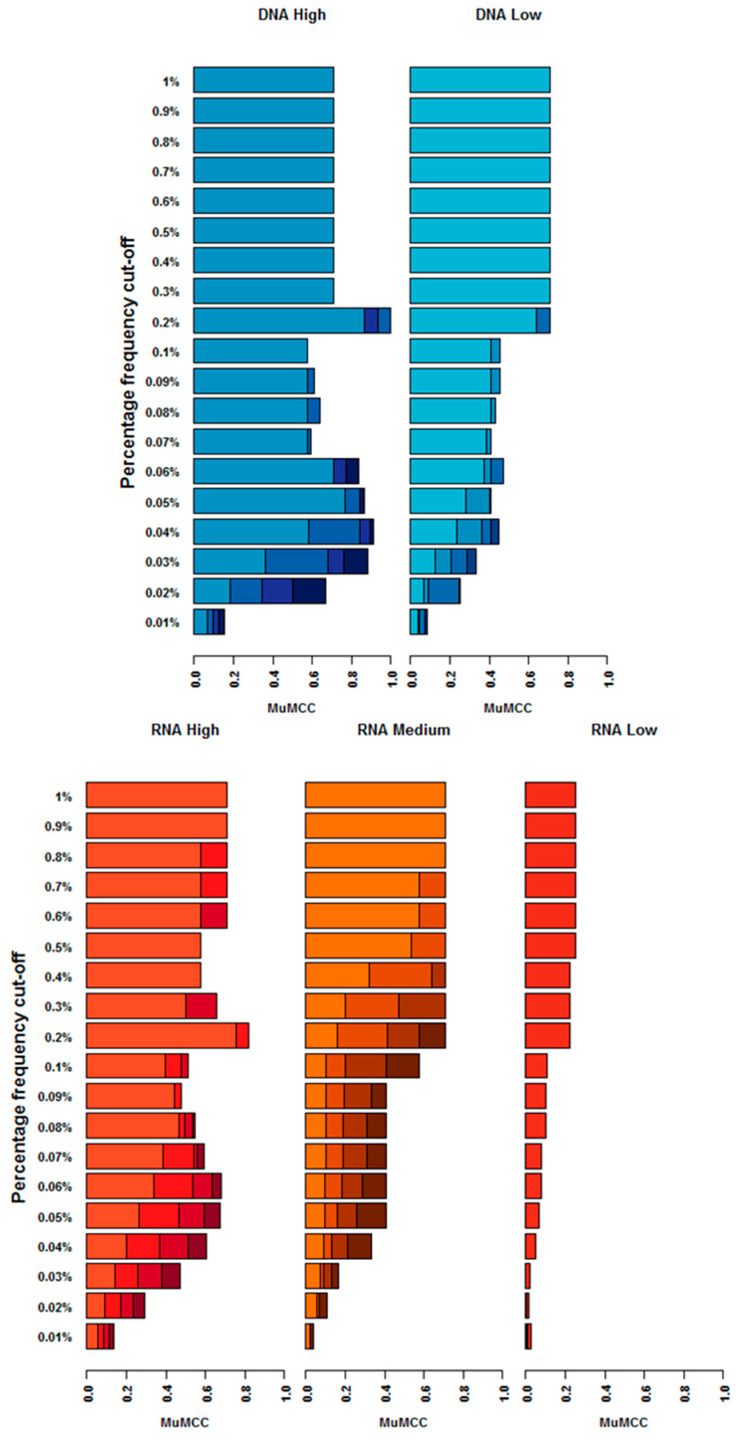
The MuMCC range for each population input for each percentage frequency cut-off parameter tested. Data obtained following alignment using GEM3 and fixed qScore and read length parameters. The data are represented is stacked, with singlet, duplicate, triplicate and quadruplicate replicate combinations for each population type represented by a different shade of colour. Darker colours represent a greater number of replicates with increased MuMCC score benefit.

**Figure 5 viruses-12-01187-f005:**
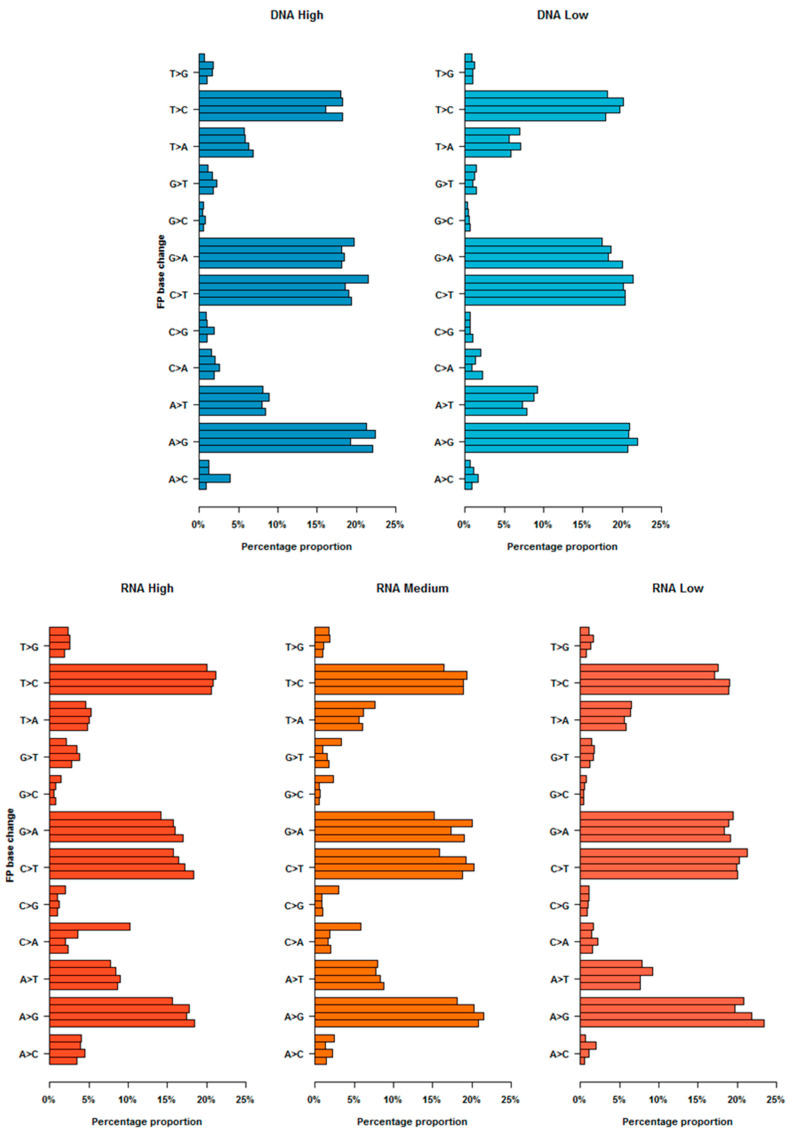
Identification of nucleotide substitutions which arose from processed-introduced error. All error base changes (from all frequency cut-offs tested) from each sequenced technical replicate (represent by each of the 4 bar plots in each base change) for each population type were analysed. Regardless of genomic type and abundance, four main False Positive (FP) base changes (represented >10% of all FP types) were characterized in each of the four technical replicates.

**Table 1 viruses-12-01187-t001:** The location of the site-directed mutations introduced in plasmids pT7S3 A–D compared with the original pT7S3 wild type. The relative abundance of each used to create the artificial populations is provided along with the resulting single-nucleotide variant (SNV) percentage frequency*. Amplicon position is defined as the number of bases along the polymerase chain reaction (PCR) amplicon.

		Original	Site-Directed Mutagenesis							
		pT7S3 Wild Type	pT7S3 A	pT7S3 B	pT7S3 C	pT7S3 D	Nucleotide Frequency			
Plasmid Relative Abundance		0.01%	1.00%	10.00%	88.89%	0.10%	A	T	C	G
Amplicon position*	1754	C	T	T	T	C		99.89%	0.11%	
	1932	G	G	G	G	A	0.10%			99.90%
	2149	G	A	A	G	G	11.00%			89.00%
	2297	T	G	G	G	G		0.01%		99.99%
	2323	A	G	G	G	G	0.01%			99.99%
	2505	A	A	G	G	A	1.11%			98.89%
	2507	T	T	G	G	T		1.11%		98.89%
	2755	G	A	A	A	A	99.99%			0.01%
	2761	A	T	T	T	A	0.11%	99.89%		
	2767	G	A	A	A	A	99.99%			0.01%
	2791	C	T	T	T	T		99.99%	0.01%	
	2843	A	C	C	C	C	0.01%		99.99%	
	2955	G	A	A	A	A	99.99%			0.01%
	3106	G	A	A	A	A	99.99%			0.01%
	3376	C	A	C	A	C	89.89%		10.11%	
	3645	G	G	G	G	A	0.10%			99.90%
	3661	G	A	A	A	G	99.89%			0.11%
	3691	T	G	G	G	T		0.11%		99.89%
	3695	G	T	T	T	G		99.89%		0.11%
	3697	T	C	C	C	T		0.11%	99.89%	

**Table 2 viruses-12-01187-t002:** The optimized computational parameters from all populations for low-frequency SNV characterisation. A list of chosen aligner and qScore, read length and frequency thresholds which produced the highest SNV detection accuracy for all population types and replicate numbers is given. The RNA *Low* input was removed due to its inability to accurately call low-frequency SNVs.

	Input	Replicate Combinations	Aligner	qScore	Read Length (bp)	Suggested Frequency Cut-Off	MuMCC
RNA	*High*	Singlet	GEM3	38	70	0.20%	0.756
		Duplicate				0.20%	0.816
		Triplicate				0.20%	0.816
		Quadruplicate				0.20%	0.816
	*Medium*	Singlet	GEM3	38	70	0.80%	0.707
		Duplicate				0.50%	0.707
		Triplicate				0.30%	0.707
		Quadruplicate				0.20%	0.707
DNA	*High*	Singlet	GEM3	35	70	0.20%	1.000
		Duplicate				0.20%	0.933
		Triplicate				0.04%	0.891
		Quadruplicate				0.04%	0.913
	*Low*	Singlet	GEM3	35	70	0.20%	0.707
		Duplicate				0.20%	0.707
		Triplicate				0.20%	0.707
		Quadruplicate				0.20%	0.707

**Table 3 viruses-12-01187-t003:** The percentage frequency and base change of each FP identified above the recommended frequency threshold for each population input and replicate. For the RNA *Low* input, a suggested frequency cut-off was not identified.

				Input Replicate
Input	Suggested Frequency Cut-Off	Amplicon Position	Base Change	1	2	3	4
DNA *Low*	0.2%	605	G > T				0.22%
RNA *High*	0.2%	1135	T > C	0.39%	0.35%	0.41%	0.30%
0.2%	1915	T > C		0.21%		0.28%
0.2%	2300	T > C			0.21%	
0.2%	3056	T > C	1.06%	0.81%	0.62%	0.87%
RNA *Medium*	0.8%	1135	T > C	1.14%			
RNA *Low*		1609	T > C		53.76%		
	2199	G > T			18.49%	
	2744	G > C				99.94%
	2933	T > C	9.06%			
	3648	T > C		45.06%

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
