# Peer review of "A Systematic Evaluation of High-Throughput Sequencing Approaches to Identify Low-Frequency Single Nucleotide Variants in Viral Populations"

_viruses, 2020, doi:10.3390/v12101187_

Round 1
Reviewer 1 Report
This is a very good study which should be published. However, there are a few issues that can improve its quality and make it easier for readers to understand:
- In the abstract and title of the article, the authors say that they evaluate low-frequency variants with high-throughput sequencing. But all of their experiments were done on Illumina. However, there are different approaches to high-throughput sequencing. There are different methods for library preparation and different sequencers have their own characteristics. All this significantly affects the result. Therefore, the authors should, at least in the abstract, indicate that the work was done using MiSeq.
- Supplementary table 1. Give a more detailed description of the table. What does the “-” mean? Is this a deletion?
- In supplementary table 3, the authors provided information on how many reads (total reads) were spent on the sample. However, there is nowhere (I may have missed) the data on what coverage was obtained for the samples analyzed (min, max, mean, median). It will be useful if the authors add this data to the supplementary table 3.
Reviewer 2 Report
This manuscript describes a systematic evaluation of various factors impacting the accuracy of variant calling in viral populations using high-throughput sequencing. Identifying variation within a viral population is important for understanding viral evolution and transmission, but this process involves various laboratory and bioinformatic steps which all affect the final outcome. Through experiments on artificially constructed viral populations, the authors provide valuable insights regarding the design of a variant calling workflow. The manuscript is well written and in general easy to follow.
Broad comments:
- My main concern regarding the manuscript as it stands is that the authors speak of variant calling, without mentioning any variant calling software. Since no variant calling algorithm is described anywhere, my conclusion is that the authors determine variants based on a very straightforward approach, simply by counting read alignments per nucleotide per position. While this is a valid method, more sophisticated algorithms have been developed over the last decade (e.g. V-Phaser2, VirVarSeq, Quasitools, LoFreq, to name a few). These methods take more information into account, such as alignment quality, base quality, coverage, etc, and are therefore expected to enable improved variant calling (in particular in terms of false positives). In a manuscript like this, when talking about variant calling, one should evaluate performance of these specialized tools. It would be very interesting to see how these compare, also to the straightforward counting approach used by the authors.
- Another comment regarding the variant calling principle, is that this manuscript considers only SNPs. It would be appropriate to refer to SNP calling rather than variant calling throughout the manuscript, to emphasize that no other types of polymorphisms are considered.
- The introduction gives a brief description of the topic, its relevance and related work, but misses the concept of variant calling as well as haplotype reconstruction (a related approach to identify variation within a population—see for example: Cacciabue, Marco et al. “A beginner's guide for FMDV quasispecies analysis: sub-consensus variant detection and haplotype reconstruction using next-generation sequencing.” Briefings in bioinformatics, bbz086. 4 Nov. 2019, doi:10.1093/bib/bbz086).
- The methods section describes how the artificial populations were constructed, but is very unclear about the final composition of the populations. Table 1 presents the relative abundances of all substitutions, but it wasn’t until I saw Supplementary Table 1 that I understood the actual composition of the strains. This should be better explained in the main text, for example by mentioning the relative frequencies of each of the plasmids in the final population (as in Supplementary table 1). But even after seeing the supplementary material I was still confused, because the numbers from Sup. Table 1 don’t match the numbers in Table 1: for example, the first variant (pos 1754) occurs in strains A, B and C, adding to a total abundance of 63.9+25.0+10.0=98.9%, while Table 1 states a frequency of 99.99%. Also the next variant doesn’t add up, so it looks like something is quite wrong with these tables.
- The authors use the MCC measure to evaluate variant calling performance, which seems an appropriate choice, but I’m confused by the mixed use of the terms MCC and MuMCC. MuMCC is defined in section 2.4.1, but MCC is not defined anywhere. If the authors intentionally distinguish between MCC and MuMCC, then MCC should also be defined in section 2.4.1. If not, please stick to one notation.
- Section 3.3 discusses the choice of qScore on the accuracy of variant calling, but it is unclear how this threshold is applied. From the supplementary material, I came to understand that this is the parameter used as input for read trimming with Sickle? That would mean that any bases with a base quality below qScore are trimmed from the read ends? This should be clearly stated in Section 3.3, otherwise the reader cannot understand the meaning of this parameter.
- I found Section 3.5 and Figure 4 very hard to interpret. What does all of this mean? Instead of evaluating these frequency cut-offs, it makes more sense to evaluate performance of different combinations of read mappers and variant callers. Then one could subsequently select the optimal combination and evaluate performance as a function of variant frequency.
- In the final conclusions, the authors write that for low RNA inputs “the use of more than one technical replicate decreased variant MCC score to 0, indicating that variants called were no better than random”. I would guess that the problem with combining technical replicates for low RNA inputs is that hardly any variant calls can be made, because there is insufficient coverage for replicates to share many variants. But I would expect any calls that can be made from replicates to be highly accurate, and if so, these are certainly better than random.
Specific comments:
- L25: “repetitive error patterns”. I thought this meant the authors discovered specific repeat sequences causing trouble, but all I could find in the manuscript is the conclusion to be careful with specific base substitutions at low frequencies. Unless I missed something, this sentence should be reformulated to clarify this.
- L35: “understanding viral sequence” -> understanding viral populations
- L44: “any bias introduced at either step will potentially be present” -> any technical errors produced at either step will be indistinguishable from true variation
- L71: “reproducible patterns of error”. I find the term “reproducible” very confusing here: do you mean that if you were to repeat the entire workflow from scratch, you would find the exact same errors? I think “systematic biases” would be more appropriate here.
- L166: “reference sequencing” -> reference genome
- L175: “a variant at a non-variant site” -> a variant called at a non-variant site
- L177: “The calling of a variant was dependent on the frequency threshold being applied”. Does this mean variants below the threshold were considered as TN irrespective of the actual base call, i.e. also correct calls become TN? Please clarify.
- L201: “the parameters that produced”. Which parameters?
- L203: “reproducible error”. See comment for L71 above.
- L240: “significant reduction” -> significant increase (?!)
- Fig 1: boxes on different levels have different shapes, but there doesn’t seem to be any meaning to the shapes used. Why do different levels use different shapes? I found it a little confusing and distracting.
- Table 1 caption: “percentage” -> relative abundance
Round 2
Reviewer 2 Report
I would like to thank the authors for addressing my comments satisfactorily. Although I would still be interested to see how different variant callers perform on the datasets presented in this paper, I agree that the manuscript as it stands is of high interest to the viral sequencing community.